

# An integrated model of seasonal changes in stock composition and abundance with an application to Chinook salmon

Cameron Freshwater[1], Sean C. Anderson[1,2], Terry D. Beacham[1],
Wilf Luedke[3], Catarina Wor[1] and Jackie King[1]

[1] Pacific Biological Station, Fisheries and Oceans Canada, Nanaimo, British Columbia, Canada
[2] Department of Mathematics, Simon Fraser University, Burnaby, British Columbia, Canada
[3] South Coast Area Office, Fisheries and Oceans Canada, Nanaimo, British Columbia, Canada

Corresponding author
Cameron Freshwater,
cameron.freshwater@dfo-mpo.gc.ca

## ABSTRACT

Population-specific spatial and temporal distribution data are necessary to identify mechanisms regulating abundance and to manage anthropogenic impacts. However the distributions of highly migratory species are often difficult to resolve, particularly when multiple populations' movements overlap. Here we present an integrated model to estimate spatially-stratified, seasonal trends in abundance and population composition, using data from extensive genetic sampling of commercial and recreational Chinook salmon (*Oncorhynchus tshawytscha*) fisheries in southern British Columbia. We use the model to estimate seasonal changes in population-specific standardized catch per unit effort (a proxy for abundance) across six marine regions, while accounting for annual variability in sampling effort and uncertain genetic stock assignment. We also share this model as an R package stockseasonr for application to other regions and species. Even at the relatively small spatial scales considered here, we found that patterns in seasonal abundance differed among regions and stocks. While certain locations were clearly migratory corridors, regions within the Salish Sea exhibited diverse, and often weak, seasonal patterns in abundance, emphasizing that they are important, year-round foraging habitats. Furthermore, we found evidence that stocks with similar freshwater life histories and adult run timing, as well as relatively proximate spawning locations, exhibited divergent distributions. Our findings highlight subtle, but important differences in how adult Chinook salmon use marine habitats. Down-scaled model outputs could be used to inform ecosystem-based management efforts by resolving the degree to which salmon overlap with other species of concern, as well as specific fisheries. More broadly, variation in stock-specific abundance among regions indicates efforts to identify mechanisms driving changes in size-at-maturity and natural mortality should account for distinct marine distributions.

## INTRODUCTION

Spatial and temporal distributions determine exposure to physical and biological processes that regulate population dynamics. Distribution data are critical when managing exploited species, where harvest impacts must be disentangled from natural drivers (*Hilborn & Walters, 1992*), and in migratory species, where long-distance movements may span multiple political jurisdictions and function as bottlenecks in survival (*Runge et al., 2014*; *Secor, 2015*). These complications are particularly relevant to Pacific salmon (*Oncorhynchus* spp.), since they support large-scale fisheries and undergo extensive migrations through freshwater and marine ecosystems. Juvenile Pacific salmon migrate to the ocean and disperse along the continental shelf or offshore to mature, returning to spawn from several months to six years later. Since Pacific salmon typically return to their natal streams, freshwater migrations are well described at the population level and variation among stocks has long informed fisheries management. However, population-specific marine migrations, particularly at fine temporal scales, are often more poorly resolved due to the long distances that are travelled and extensive overlap among populations (*Quinn, 2018*). Marine distributions of Pacific salmon are of interest because substantial mortality (*Parker, 1968*; *Seitz et al., 2019*), as well as the vast majority of somatic growth (*Quinn, 2018*), occurs during marine residence. Thus knowledge of marine distributions, which regulate exposure to bottom-up processes, predators, and fisheries, is necessary to identify mechanisms responsible for changes in productivity.

Unlike most other North American Pacific salmon species, Chinook salmon (*O. tshawytscha*) exhibit a mix of nearshore and offshore distributions and may be harvested year-round in mixed stock marine fisheries (*Riddell et al., 2018*). Since the 1970s, large-scale tagging programs have been used to estimate Chinook salmon harvest rates and distributions to inform management decisions. Most commonly, juvenile fish are tagged with coded wire tags (CWTs) that identify an individual to a release group and which are recovered in fisheries or on spawning grounds. The abundance and age-at-maturity of a tagged stock can be estimated by assuming age-specific natural mortality rates, as well as applying expansions to account for sampling effort (*Johnson, 2004*; *Nandor, Longwill & Webb, 2010*). Widespread tagging was originally intended to inform multinational management negotiations (such as the Pacific Salmon Treaty); however, these data have also provided substantial information on juvenile (*Trudel et al., 2009*; *Fisher et al., 2014*) and adult (*Weitkamp, 2010*; *Shelton et al., 2019*) marine distributions. For example, CWT studies provided evidence of differential migration patterns among Chinook salmon life history types (*Fisher et al., 2014*), demonstrated stocks often exhibit regionally coherent marine distributions (*Weitkamp, 2010*), and revealed stock-specific responses to climate change impacts (*Shelton et al., in press*).

CWT recoveries, however, provide an imperfect estimate of Pacific salmon distributions. A subset of stocks serve as indicators for larger stock groups and, due to convenience, these indicators are often hatchery populations. Although direct comparisons are limited, there is evidence that the distribution of some Pacific salmon stocks may not be well represented by recoveries of their indicators (*Winther & Beacham,*

*2006*; *Peterson, Clark & Evenson, 2016*; *Beacham et al., 2019*). Furthermore, salmon enhancement programs have changed over time, with indicator groups discontinued or added, creating gaps in the time series of life history types that are of particular management concern (e.g., Upper Fraser River yearling spring run Chinook salmon; *DFO, 2020*) and complicating efforts to evaluate interannual changes in stock composition. The implementation of mass marking strategies that consist of large releases of marked and untagged (i.e., adipose fin clipped and no CWT) individuals, coupled with mark-selective fisheries has reduced the efficiency of CWT recovery programs (*PSC, 2005*). As a result, estimating contemporary stock-specific distributions, especially at fine spatial or temporal scales, is not always feasible using CWT recoveries alone.

Natural tags, most commonly genetic stock identification (GSI) techniques, have increasingly been incorporated into Pacific salmon management frameworks (*Shaklee et al., 1999*; *Dann et al., 2013*). GSI can be used to identify individual fish to their population of origin using microsatellites or single nucleotide polymorphisms and is the only means of reliably identifying stock of origin when an individual is not tagged by a management agency. Furthermore, GSI allows a greater proportion of the sample to inform composition estimates because all fish are "tagged". GSI is commonly used to estimate the stock composition of fisheries-independent surveys (*Tucker et al., 2012*), as well as escapement, bycatch, and terminal fisheries (*Wilmot et al., 1998*; *Shaklee et al., 1999*; *Hess et al., 2014*). More recently, GSI has begun to be incorporated into the management of more highly mixed open-ocean fisheries (*Satterthwaite et al., 2014*), where it has been used, often in tandem with CWT data, to inform time-area closures (*Winther & Beacham, 2006*; *Dobson, Holt & Davis, 2020*).

Mixed-stock fishery challenges are particularly acute in southern British Columbia. The region is used by a diverse assemblage of Chinook salmon stocks (*Weitkamp, 2010*), including populations spawning as far south as central California. While some stocks encountered by southern BC fisheries are above their management reference points, many others are at low abundance and several Canadian-origin stocks are of conservation concern (*COSEWIC, 2018*). Furthermore, declines in Chinook salmon abundance have co-occurred with persistently low population growth rates for southern resident killer whales (*Orcinus orca*; *Ward, Holmes & Balcomb, 2009*; *Vélez-Espino et al., 2015*). Southern resident killer whales appear to prey heavily upon Fraser River Chinook salmon (*Hanson et al., 2010*), several populations of which are currently at low abundance (*COSEWIC, 2018*), creating ecosystem-based fisheries management incentives to promote Chinook salmon recovery. Finally, there is growing evidence that generic categories of Chinook salmon marine distributions, such as offshore migrants or continental shelf residents, fail to capture subtle differences in habitat use. For example, Puget Sound Chinook salmon appear to exhibit partial residency where considerable portions of certain populations remain within that basin (*O'Neill & West, 2009*; *Chamberlin et al., 2011*).

To address these challenges, biologists and fisheries managers in BC have used recovery of anthropogenic tags (CWTs and thermally marked otoliths), as well as GSI to refine time-area closures that minimize impacts on stocks of concern. For example, intensive

sampling of fisheries near Haida Gwaii and west coast Vancouver Island (WCVI) revealed that depleted, wild WCVI populations disproportionately use nearshore migration corridors (*DFO, 2012*; *Winther & Beacham, 2006*). Similar sampling of fisheries throughout southern BC has been used to resolve fine-scale marine migration patterns of early run Fraser River stocks (*Dobson, Holt & Davis, 2020*). In both cases fishery closures have resulted in reduced harvest of at-risk stock groups (*Beacham et al., 2008*; *Dobson, Holt & Davis, 2020*). However, such data are applied on a case-by-case basis and have not yet been synthesized to generate predictions of stock-specific marine distributions or relative abundance throughout BC. Similarly, previous examinations of adult Chinook salmon distributions have focused on regions with lower stock diversity (*Satterthwaite et al., 2013*; *Satterthwaite et al., 2015*; *Bellinger et al., 2015*) or relatively coarse ecological, spatial, and temporal scales (*Weitkamp, 2010*; *Larson et al., 2013*; *Shelton et al., 2019*).

We build on these findings by presenting estimates of seasonal changes in the distribution of adult Chinook salmon in southern BC derived from extensive GSI sampling of commercial and recreational fisheries. We develop a flexible, integrated model, which accounts for uncertainty in individual genetic stock assignments and uses splines to generate smoothed predictions over the annual cycle. We focus our analysis on regional stock aggregates that describe general patterns associated with spawning regions and life-history type, as well as Canadian-origin stocks relevant to domestic management actions. Although our estimates are generated at relatively large spatial scales to maximize seasonal coverage, the model can be down-scaled to generate predictions at finer ecological and spatio-temporal scales and is available within our included R package stockseasonr. Ultimately spatially and temporally explicit estimates of stock composition and relative abundance can improve our understanding of how Chinook salmon stocks use distinct nearshore habitats and differ in exposure to drivers of population dynamics.

## METHODS

### Study system

Chinook salmon are harvested by American and Canadian commercial, recreational, and First Nations fisheries, prior to and during return migrations to freshwater spawning habitats. In Canada, management decisions are often applied at the scale of DFO's Pacific Fishery Management Areas (PFMAs) (*DFO, 2018*). A PFMA may contain multiple subareas, with a PFMA denoted by a numeric and a subarea by an alphabetical. Our analysis focused on PFMAs throughout southern British Columbia (i.e., the west coast of Vancouver Island (WCVI) and the Canadian portions of the Salish Sea), which we aggregated into six catch regions based on proximity and shared oceanographic features (Fig. 1). Note that we moved two PFMA subareas (13M and 13N), located in the northern Strait of Georgia, to that catch region from the Queen Charlotte/Johnstone Strait region. The timing and location of fisheries restrictions within a given PFMA may change to avoid stocks of concern, as well as changes in quota as determined under the Pacific Salmon Treaty. For example, in recent years, management actions have resulted in the commercial troll fishery (largely restricted to outside portions of WCVI) shifting from harvesting in the late fall through early spring to harvesting only in late summer.

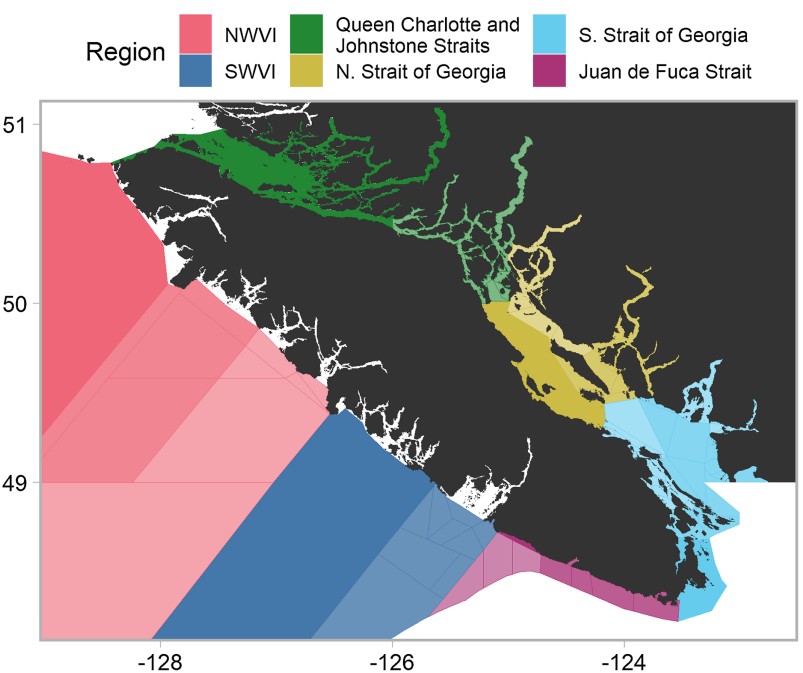

**Figure 1 Southern British Columbia catch regions and Pacific fishery management areas (PFMAs).** Shading and lines denote PFMAs and subareas within a given catch region. Unshaded portions represent areas in which insufficient data were available to estimate model parameters or areas outside Canadian jurisdiction. Commercial data originated from outside catch regions (i.e., NWVI and SWVI), while recreational data originated from inside catch regions (i.e., Queen Charlotte/Johnstone Strait, Juan de Fuca Strait, and northern and southern Strait of Georgia).

The WCVI and inside (i.e., Canadian Salish Sea plus Queen Charlotte Strait) sport fisheries operate year-round with area-specific retention regulations, but the majority of effort occurs from early summer to early fall.

## Data collection

Tissue samples for genetic stock identification (GSI) were collected from commercial and recreational fisheries by two independent sampling programs. Genetic stock assignments were performed using cBAYES and a reference baseline derived from microsatellite markers that consisted of 268-populations with more than 50,000 individuals (*Beacham et al., 2006*).

Genetic samples were collected from the commercial troll fishery, predominantly dockside and at processing facilities, from 2007–2015. Sampling was performed by observers contracted by Fishery and Ocean Canada's (DFO) Mark Recovery Program to recover CWTs. The Mark Recovery Program aims to sample 20% of the landed WCVI commercial catch and these samples were further sub-sampled to provide GSI samples with a target of 4% of the monthly catch. We note that GSI samples from the commercial fishery could be attributed to a catch region and landing day, but not to a specific spatial location or harvest date because trollers may fish multiple PFMAs and remain at sea for several days before landing their catch. Additional GSI samples originated from two other WCVI fisheries. The first was a contracted test fishing troller with an at-sea observer,

which gathered samples in May, June, and September of 2008–2011 from locations in the NWVI catch region immediately before and after the standard commercial opening. The second was the T'aaq-wiihak fishery, a First Nations economic-opportunity fishery, that operates on the west coast of Vancouver Island. T'aaq-wiihak fishery openings may occur at different times than standard commercial fisheries and we only retained samples in this analysis that overlapped with commercial openings. Commercial catch (individual fish) and effort (boat days) data were retrieved from DFO's Fisheries Operations System database for all PFMAs and years in which GSI samples were available. These data consist of daily individual catch values from mandatory vessel logbooks, as well as the number of licensed vessels operating on a given day in each PFMA. Data from the commercial fishery were restricted to the northwest Vancouver Island (NWVI) and southwest Vancouver Island (SWVI), otherwise referred to as "outside", catch regions (Fig. 1). We included commercial catch and effort data from 2007–2015 with catch and composition data available for all months except July in SWVI (18,470 individual samples; Figs. S1; S2).

Genetic samples were collected from the recreational fishery by dockside creel survey observers, as well as via a citizen science program, Avid Anglers, which is a collaboration between DFO, recreational harvesters, sport fishing guides, and the Pacific Salmon Foundation. The program encourages individuals who are consistent fishers (e.g., guides) to provide size and location data for all salmon that they encounter (kept and released), as well as sampling at least one fish per day for GSI. We aggregated the recreational GSI data to catch region and month to ensure adequate sample sizes and facilitate comparisons with commercial data. Recreational catch data were obtained from DFO's creel database, which records estimates of monthly catch (individual fish) and effort (boat days) at the subarea level based on dockside fisher interviews and regular fly-overs (English, Searing & Nagtegaal, 2002). Sufficient GSI samples and catch data from recreational fisheries were only available from PFMAs within "inside" regions (i.e., Queen Charlotte and Johnstone Strait, the Strait of Georgia, and Juan de Fuca Strait; Fig. 1) and, with the exception of southern Strait of Georgia GSI data, were only available for a subset of months. Therefore, we fit the recreational models to data from these catch regions for months between January and December, with specific ranges differing among regions (Fig. S2). Recreational catch/effort data spanned 2009–2019 and stock composition data (11,729 individual samples) spanned 2014–2019 (Figs. S1; S2). All biological sampling was covered by a blanket Section 52 license relevant to Fisheries and Oceans Canada field activities for management purposes.

We pooled populations with similar freshwater distributions and run timing to generate two sets of stock aggregates. The first grouping (Regional, Table 1) contained aggregates based on those defined in Pacific Salmon Treaty documents (CTC, 2019), with the modification that several regional aggregates that are less common in southern BC fisheries were pooled and that Columbia River stocks were grouped based on common juvenile patterns of marine dispersal (Fisher et al., 2014). The second, Canadian-centric grouping retained fine-scale Canadian-origin aggregates most relevant to southern BC management

**Table 1 Regional and Candian-centric stock aggregates, as well as associated abbreviations.** Pooled groupings (i.e., CA/OR-coast and NBC/SEAK) include stock aggregates that are relatively rare in southern BC fisheries. Columbia River Upper Spring includes Snake River individuals.

| Regional | Canadian-centric |
| --- | --- |
| California and Oregon Coastal (CA/OR-coast) | Other |
| Columbia River Upper Spring (CR-upper_sp) | |
| Columbia River Lower Spring (CR-lower_sp) | |
| Columbia River Upper Summer/Fall (CR-upper_su/fa) | |
| Columbia River Lower Fall (CR-lower_fa) | |
| Washington Coastal (WA-coast) | |
| Puget Sound (PSD) | |
| Northern British Columbia and Southeast Alaska (NBC/SEAK) | |
| Strait of Georgia (SOG) | East-coast Vancouver Island (ECVI) |
| | Southern British Columbia Mainland (SOMN) |
| Fraser River Early Run (FR-early) | Fraser Spring 4.2 |
| | Fraser Spring 5.2 |
| | Fraser Summer 4.1 |
| | Fraser Summer 5.2 |
| Fraser Late Run (FR-late) | Fraser Fall |
| West-coast Vancouver Island (WCVI) | West-coast Vancouver Island (WCVI) |

decisions and pooled all other aggregates (Table 1). Although the PST Chinook technical committee recently included higher resolution stock groupings for certain Canadian stocks (CTC, 2020), we use the older regional groupings to improve readability and because the relevant stocks are shown in the Canadian-centric results. Our groupings largely pool stocks with evidence of similar marine distributions based on CWT recoveries (Weitkamp, 2010); however, aggregating stocks will necessarily obscure population-specific distributions. We recommend using down-scaled model versions when a small number of stocks are of interest. Additionally, we note that most stock aggregates contain a mix of hatchery- and wild-origin populations; however, the relative proportion of each varies among aggregates, as well as years, and we did not attempt to distinguish between patterns in hatchery and wild abundance.

## Stock-specific distribution model

We assumed catch per unit effort (CPUE) could be used as a proxy for relative abundance in a given spatio-temporal strata (here catch region and month), though we note that hyperstable catch rates can violate this assumption (Harley, Myers & Dunn, 2001). To predict standardized CPUE (i.e., catch of all stocks given a fixed mean effort), we modeled catch (individual fish) $C$ as a negative binomial process, with mean $\mu$ and inverse dispersion $\phi$ ($\mathrm{Var}[C] = \mu + \mu^2/\phi$), via a log link and a generalized additive model (GAM). We included log effort as an offset (i.e., fixed the effort coefficient at one), and allowed

changes in seasonal abundance to be described by splines. The GAM followed the general form:

$$C_i \sim \text{NegBin}(\mu_i, \phi), \tag{1a}$$

$$\log(\mu_i) = \alpha_y + \alpha_p + \sum_{m=1}^{M} f_{m_p}(m_i) + \log(b_i), \tag{1b}$$

$$\alpha_y \sim \left( \begin{array}{ll} \text{Normal}\left(\mu_{\alpha_y}, \sigma_{\alpha_y}^2\right), & \text{if } y = 1 \\ \text{Normal}\left(\alpha_{y-1}, \sigma_{\alpha_y}^2\right), & \text{if } y > 1 \end{array} \right. \tag{1c}$$

where $i$ is an observation, $\alpha_y$ is a random intercept for year $y$ modeled as a random walk, $\alpha_p$ is a fixed intercept for PFMA $p$, $f_m$ is a PFMA-specific smooth function for month $m$, and $b$ is the number of boat days (representing effort). Each smoother is represented by a sum of $k$ basis functions, multiplied by corresponding coefficients (*Wood, 2011*). When data were available for the full annual cycle (e.g., commercial and southern Strait of Georgia composition data) we fixed $k$ at four and fit the model with cubic cyclic splines, which constrained estimates for the first and last months of the year to converge on one another (*Wood, 2006*). When data were available for a fraction of the annual cycle, we fixed $k$ at three and fit the model using thin plate splines. Values of $k$ were chosen after preliminary model runs indicated higher values failed to converge or resulted in unrealistic seasonal patterns.

To facilitate comparison with the stock composition data, which could not be reliably assigned to PFMAs in the commercial fishery, we estimated $\mu_{mr}$ (monthly catch within each catch region $r$) as the sum of its component PFMAs:

$$\mu_{m_r} = \sum_{p=1}^{P} \mu_{m_p}. \tag{2}$$

We modeled stock composition as a Dirichlet-multinomial process—a compound distribution that accounts for variability in observed proportion data, similar to the approaches used by *Thorson et al. (2017)* and *Douma & Weedon (2019)*. We assumed predicted stock proportions $q$ are related to a vector of observed stock proportions $\theta$ and sample size $n$:

$$\mathbf{q} \sim \text{Multinomial}(\mathbf{\theta}, n). \tag{3}$$

Variability in stock proportions, associated with uncertainty in the assignment probabilities for individual fish, is approximated using a Dirichlet distribution described by a vector of positive parameters $\lambda$ representing its mean and variance.

A benefit of using a Dirichlet-multinomial, rather than multinomial, distribution is its ability to incorporate uncertainty in individual stock assignments (i.e., non-whole number observations of stock-specific counts in a sample), while accounting for variation among spatio-temporal strata in sampling effort. This contrasts with many GSI analyses

where threshold probabilities are used to assign individuals to a population (e.g., 75%), effectively assuming perfect identification and excluding data associated with ambiguous stock assignments.

In practice, we implemented the model by computing the integrated Dirichlet-multinomial distribution (*Thorson et al., 2017*). Since individual assignment probabilities were identified at the scale of spawning populations (i.e., sub-stock units), we first aggregated GSI data into the stocks described in Table 1 by summing, within an individual, all probabilities associated with populations belonging to stock *s*. We next defined a sampling event *j* as all the individuals collected on a given day and within a given catch region and created vector $v_j$ by summing the assignment probabilities of all sampled individuals. Thus $v_j$ has length equal to the total number of stocks *S* and sums to $n_j$ (i.e., $v_j$ is equivalent to $n_j \theta_j$). Using the gamma function within the likelihood function of the Dirichlet-multinomial allows it to be defined for non-negative (rather than whole number only) sample sizes:

$$L(\mathbf{q}, \rho | \mathbf{v}, n) = \frac{\Gamma(n+1)}{\prod_{s=1}^{S} \Gamma(v_s+1)} \frac{\Gamma(\rho)}{\Gamma(n+\rho)} \prod_{s=1}^{S} \frac{\Gamma(v_s + \rho q_s)}{\rho q_s}, \tag{4}$$

where $\gamma$ represents the gamma function and $\rho$ is a parameter representing overdispersion resulting from the Dirichlet distribution (*Thorson et al., 2017*).

We modelled seasonal patterns in stock composition similarly to Eq. (1b) using a logit link and the GAM:

$$\text{logit}(\mathbf{q}_j) = \beta_y + \beta_r + \sum_{m=1}^{M} g_{m_r}(m_j), \tag{5a}$$

$$\beta_y \sim \begin{pmatrix} \text{Normal}\left(\mu_{\beta_y}, \sigma^2_{\beta_y}\right), & \text{if } y = 1 \\ \text{Normal}\left(\beta_{y-1}, \sigma^2_{\beta_y}\right), & \text{if } y > 1 \end{pmatrix} \tag{5b}$$

where $\beta_y$ is a random intercept for year *y* modeled as a random walk, $\beta_r$ is a fixed intercept for catch region *r*, and $g_m$ is a catch region-specific smooth function for month *m*. We used the same number of basis functions and the same spline types for the stock composition component of the model as the aggregate abundance component. To ensure model convergence we replaced zero stock assignment probablities in $\theta_j$ with very small values (0.00001). Similarly, parameters associated with stock-region combinations that were never observed (CA/OR-coast and Fraser Spring 4.2 in the northern Strait of Georgia) were fixed at zero.

We used the negative binomial and Dirichlet-multinomial model components to predict seasonal changes in standardized CPUE and stock composition, respectively, in a given catch region. We first generated predictions of standardized CPUE by assuming effort was fixed at a fishery-specific (i.e., commercial or recreational) mean value. We then used the product of predicted standardized CPUE and the probability of encountering a given stock to predict standardized stock-specific CPUE, a proxy for stock-specific abundance. By integrating across random effects, these predictions can be interpreted as

representing an average year. Thus the model allows inferences to be made about the seasonal distribution of stocks in various catch regions, while accounting for interannual differences in sampling effort. Note that we also present estimates of predicted stock composition fit separately from abundance data because recreational GSI data were available for several months that lacked complete catch and effort data.

We fit the model to four datasets: regional stock aggregates captured in the commercial fishery (outside regions), regional stock aggregates captured in the sport fishery (inside regions), Canadian-origin stock aggregates captured in the commercial fishery, and Canadian-origin stock aggregates captured in the sport fishery (Table 1). We focus on regional aggregates in the main text, but include Canadian-centric results as a supplement.

We also completed two supplemental analyses to account for data collection methods that could impact our conclusions. First, we evaluated the effect of including the Avid Angler (i.e., voluntarily submitted) samples, which may be a less representative sample of stock composition, by repeating the composition analysis for the recreational fisheries, either including or excluding the Avid Angler samples. We then compared the predictions of models fit to the full and restricted datasets. Excluding the Avid Angler samples reduced the total number of samples by ~50% and this analysis was limited only to early summer to fall months. These results are presented in the main text and as supplemental figures. Second, we evaluated the effect of fisheries restrictions in Juan de Fuca Strait and the southern Strait of Georgia on abundance and stock composition estimates. Management of Chinook salmon fisheries in southern BC includes spatio-temporal closures, as well as mark- and size-selective fisheries, to reduce impacts on stocks of concern. We included GSI samples collected from released individuals in all our analyses and, in a supplemental methods section, evaluated whether these samples adequately represent the activity of the recreational fishery.

We emphasize that estimates of commercial and recreational CPUE are independent, relative proxies for abundance and their scales should not be considered equivalent for several reasons. First, each dataset was collected over different time periods. Second, estimates of recreational catch and effort are substantially less precise than commercial equivalents due to differences in reporting requirements. Third, catchability differs between the fisheries because of differences in gear type and regulations, resulting in distinct relationships between CPUE and abundance. Fourth, the parameters in each fishery's model were estimated independently and predictions were generated with effort standardized to a fishery-specific mean. Thus the "effort effect" is different and not directly comparable between commercial and recreational models.

We identified parameter values that maximized the marginal likelihood with respect to fixed effects while integrating across random effects via the the non-linear minimizer nlminb in R 4.0.2 (*R Core Team, 2020*). We first used the mgcv R package to define model matrices containing the appropriate splines to estimate seasonal and effort effects (*Wood, 2011*). We then fit the models with TMB, which implements the Laplace approximation to integrate across random effects and the generalized delta method to compute standard errors of all fixed and random effects, as well as derived quantities (*Kristensen et al., 2016*). We computed the standard errors on predictions in link space and

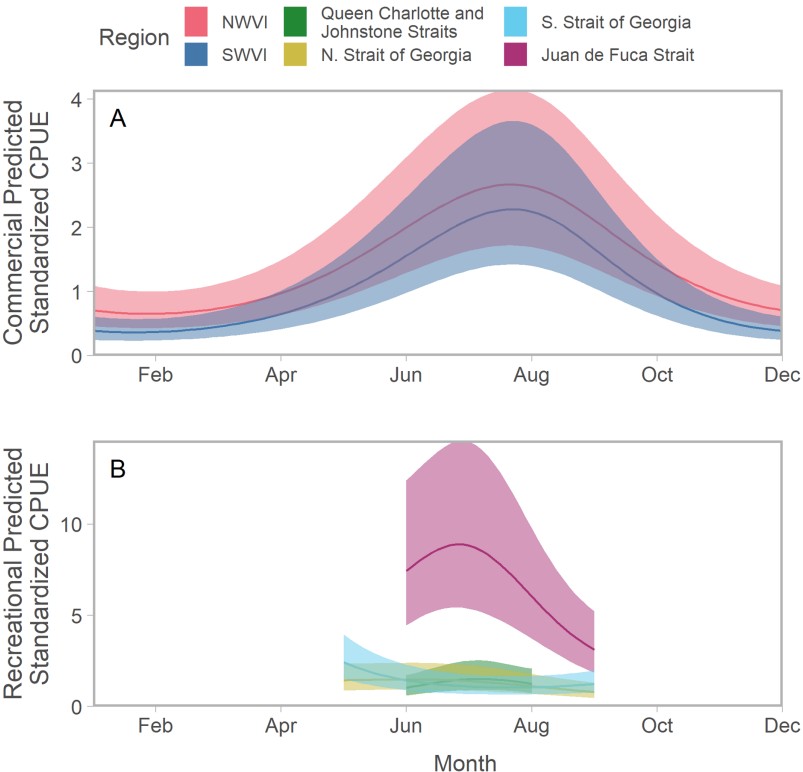

**Figure 2 Seasonal trends in model-predicted aggregate standardized catch per unit effort (represents thousands of fish assuming fixed mean effort) in southern BC catch regions (colours) estimated using commercial (A) and recreational (B) fisheries data.** Ribbons represent 95% confidence intervals. Predictions assume mean effort within a fishery and are for an average year, integrating over annual random effects. Note that y-axes differ among panels. Standardized CPUE is not directly comparable between commercial and recreational fisheries.

calculated confidence intervals as Wald confidence intervals. The TMB framework allows the model to be fit quickly and the model's structure is sufficiently flexible to accommodate a wide range of spatially stratified composition and catch data. Code to reproduce the analysis is available at https://github.com/CamFreshwater/chinDist and DOI 10.5281/zenodo.4672524, while the functions that allow fitting the integrated model to any similarly formatted dataset are available as an R package stockseasonr (DOI 10.5281/zenodo.4672540).

# RESULTS

To facilitate comparisons across time and space, all predictions assumed a fishery-specific mean effort that was fixed over the annual cycle. Predicted Chinook salmon standardized CPUE typically peaked between July and August; however, there was substantial variation among regions in the shape of seasonal trends (Figs. 2; S3; S4). Peaks in abundance were most noticeable on the west coast of Vancouver Island, where catch and effort data were available for all months (Fig. 2A). CPUE in the Strait of Georgia, particularly in the south, exhibited a different seasonal trend characterized by a weak

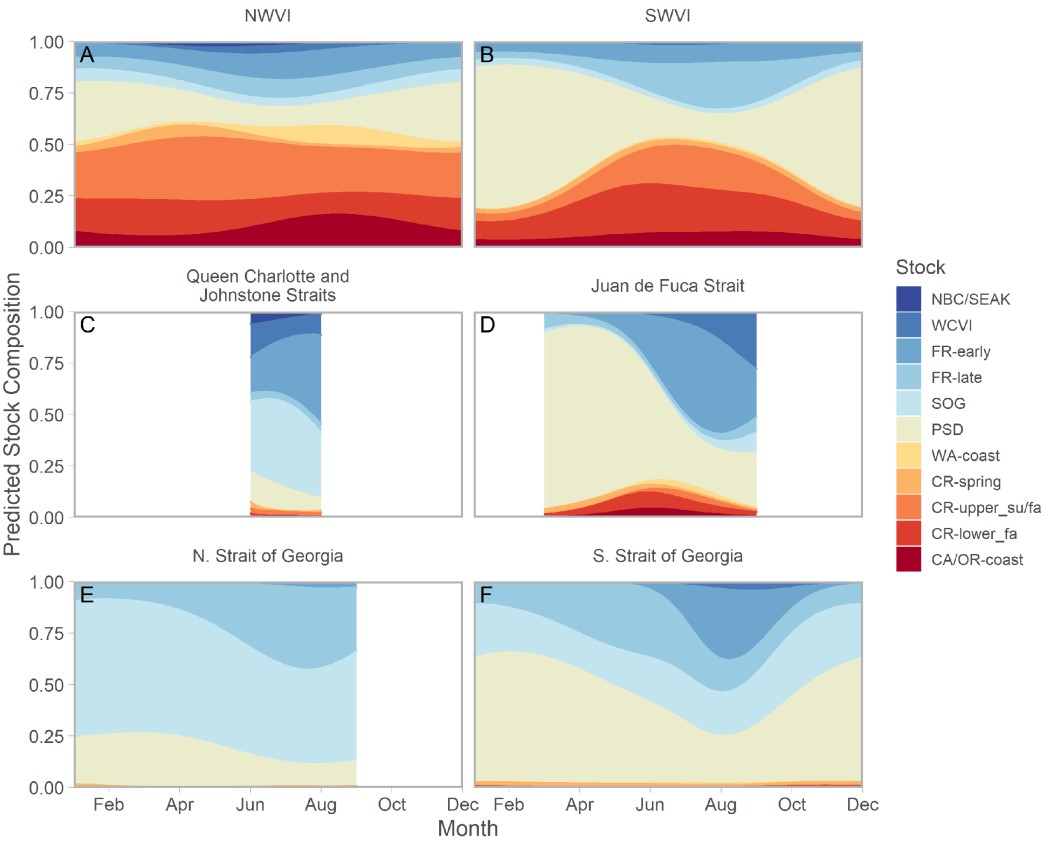

**Figure 3** Seasonal trends in model-predicted mean composition of regional stock aggregates for southern BC catch regions estimated using genetic samples collected from commercial (NWVI and SWVI; A–B) and recreational (Queen Charlotte and Johnstone straits, Juan de Fuca Strait, N. Strait of Georgia, and S. Strait of Georgia; C–F) fisheries. Stock aggregates are arranged approximately latitudinally, based on freshwater entry point, from north (blue) to south (red). To improve visibility, Columbia River upper and lower spring run stocks were pooled for this figure only. Predictions are for an average year, integrating over annual random effects. Portions of the year lacking sufficient composition data in a given region are blank.           

decline in abundance from a peak in spring (Fig. 2B). Estimates of standardized CPUE also varied substantially among years (Figs. S5; S6).

We found evidence of seasonal changes in stock composition in all regions. For example, in SWVI, Puget Sound individuals present in winter and early spring were gradually outnumbered by Columbia River upper summer/fall and lower fall run, as well as Fraser River late run stocks. Seasonal changes in NWVI were less pronounced, but relatively small contributions of Columbia upper spring, northern BC/southeast AK, and WCVI stocks were replaced by California/Oregon-coastal and Washington-coastal stocks in the late summer (Figs. 3A, 3B; S7). The Canadian-centric model indicated Fraser River Fall and, to a lesser extent, Fraser Summer 4.1 stocks were the most common Canadian-origin stocks (Figs. S8; S9).

Although GSI sampling of inside regions was more limited, there was still evidence of complex seasonal trends in composition, with the relative abundance of stocks varying as the annual cycle progressed. In the southern Strait of Georgia and Juan de Fuca Strait,

Puget Sound stocks present during the winter and spring were gradually replaced by a more balanced composition (Figs. 3D, 3F; blue and magenta in Fig. S10). In the southern Strait of Georgia, first Fraser River late run, then Fraser River early run stocks increased in relative abundance, with the contribution of local stocks (i.e., SOG, which includes east coast Vancouver Island and southern BC mainland populations) remaining relatively stable. In Juan de Fuca Strait, the contribution of Fraser River early run stocks increased in summer before declining, as the relative abundance of WCVI stocks increased. Fraser River late run stocks were notably absent (Fig. 3D; magenta in Fig. S10). Fraser River late run were relatively common in the northern Strait of Georgia, but absent in Queen Charlotte/Johnstone Straits where they were replaced by Fraser River early run stocks; however, both regions had strong contributions of SOG populations (Fig. 3C, 3E; green and gold in Fig. S10). The Canadian-centric model clarified that Fraser River early run components were made up predominantly of subyearling life histories (i.e., Fraser Summer 4.1), although Fraser Spring 5.2 populations made a substantial contribution in Juan de Fuca Strait (Figs. S8; S11).

We identified seasonal trends in stock-specific abundance by simultaneously estimating aggregate standardized CPUE, which accounted for seasonal variation in effort, and stock composition in an integrated model. On the west coast of Vancouver Island, stock aggregates fell along a continuum from year-round residence to compressed distributions peaking in late summer (Fig. 4). Puget Sound stocks were the best example of the former pattern, but in NWVI SOG and three of the four Columbia River stock aggregates were also present year-round, albeit at low abundance. Maximum abundance was typically greater in NWVI than SWVI, with the exception of Columbia River lower spring and fall run, Puget Sound, and Fraser River late run stocks. The Canadian-centric model generated qualitatively similar predictions, with evidence of strong seasonal peaks in abundance and typically greater stock-specific abundance in NWVI (Fig. S13). The major distinction when results focused on Canadian stock aggregates was that Fraser Spring 4.2 and Southern Mainland populations had 95% confidence intervals that approached zero for considerable portions of the year.

Seasonal patterns in stock-specific standardized CPUE were more variable among inside catch regions (Fig. 5). Juan de Fuca Strait exhibited strong and compressed seasonal peaks in standardized CPUE, which differed in breadth and timing among stocks (Fig. 5; magenta in Fig. S14). Seasonal trends were particularly diverse in the southern Strait of Georgia, where stock-specific standardized CPUE was low and stable (most California Current stocks), declined through the year (Puget Sound and Strait of Georgia stocks), increased modestly (Fraser early run and WCVI), or was convex (Fraser late run) (Fig. 5; S14). Unlike most other stock aggregates, the standardized CPUE of SOG populations was concentrated in Queen Charlotte/Johnstone Straits and the northern Strait of Georgia (Fig. 5; green and gold in Fig. S14).

Stock composition results from recreational fisheries did not appear to be sensitive to the inclusion of Avid Angler (i.e., voluntarily submitted) GSI samples. Predictions from a model parameterized with only data collected by DFO creel observers were qualitatively similar to those that included both observer- and Avid Angler-collected data. The most

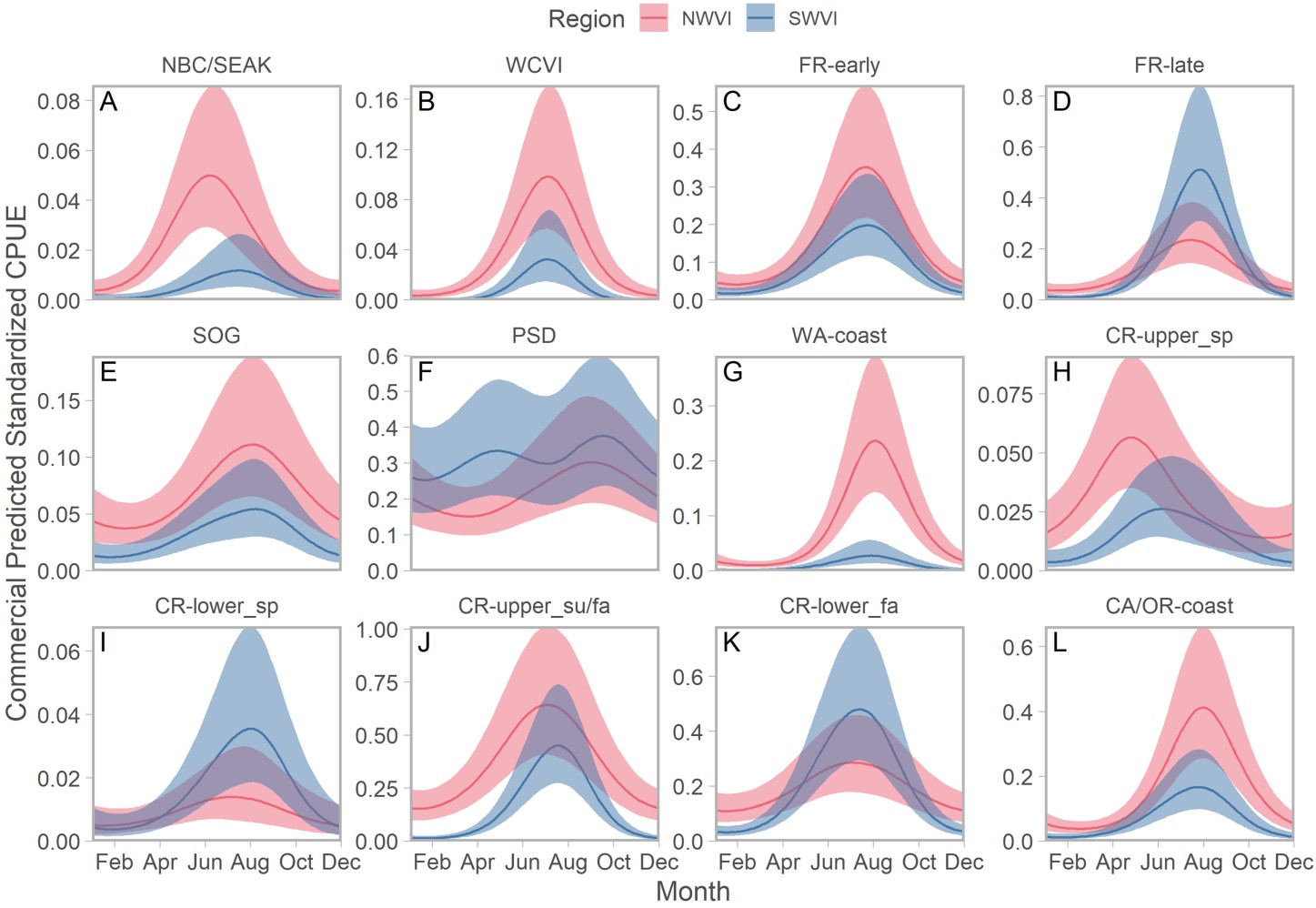

**Figure 4 Seasonal trends in model-predicted stock-specific standardized catch per unit effort (represents thousands of fish assuming fixed mean commercial effort) of regional stock aggregates in west coast Vancouver Island catch regions (colours) estimated using commercial fisheries data.** Ribbons represent 95% confidence intervals. Predictions are for an average year, integrating over annual random effects. Stock aggregates are arranged approximately latitudinally, based on freshwater entry point, from north to south (A–L). Note that *y*-axis scales differ among stock aggregates.

noticeable difference was a reduction in the contribution of Fraser River Fall stocks, replaced by Puget Sound and Fraser River Summer 4.1, in southern Strait of Georgia (Fig. S12).

Conversely, fisheries restriction in portions of Juan de Fuca Strait and the southern Strait of Georgia are likely to impact spring and early summer stock composition estimates in these regions. Since relatively few genetic samples were collected from released fish, it is likely that the relative abundance of Fraser River early run stocks were underestimated for these spatio-temporal strata (discussed in detail in Supporting Information).

# DISCUSSION

Chinook salmon marine distributions are typically classified as falling within one of several broad categories associated with freshwater life history and ocean entry location.

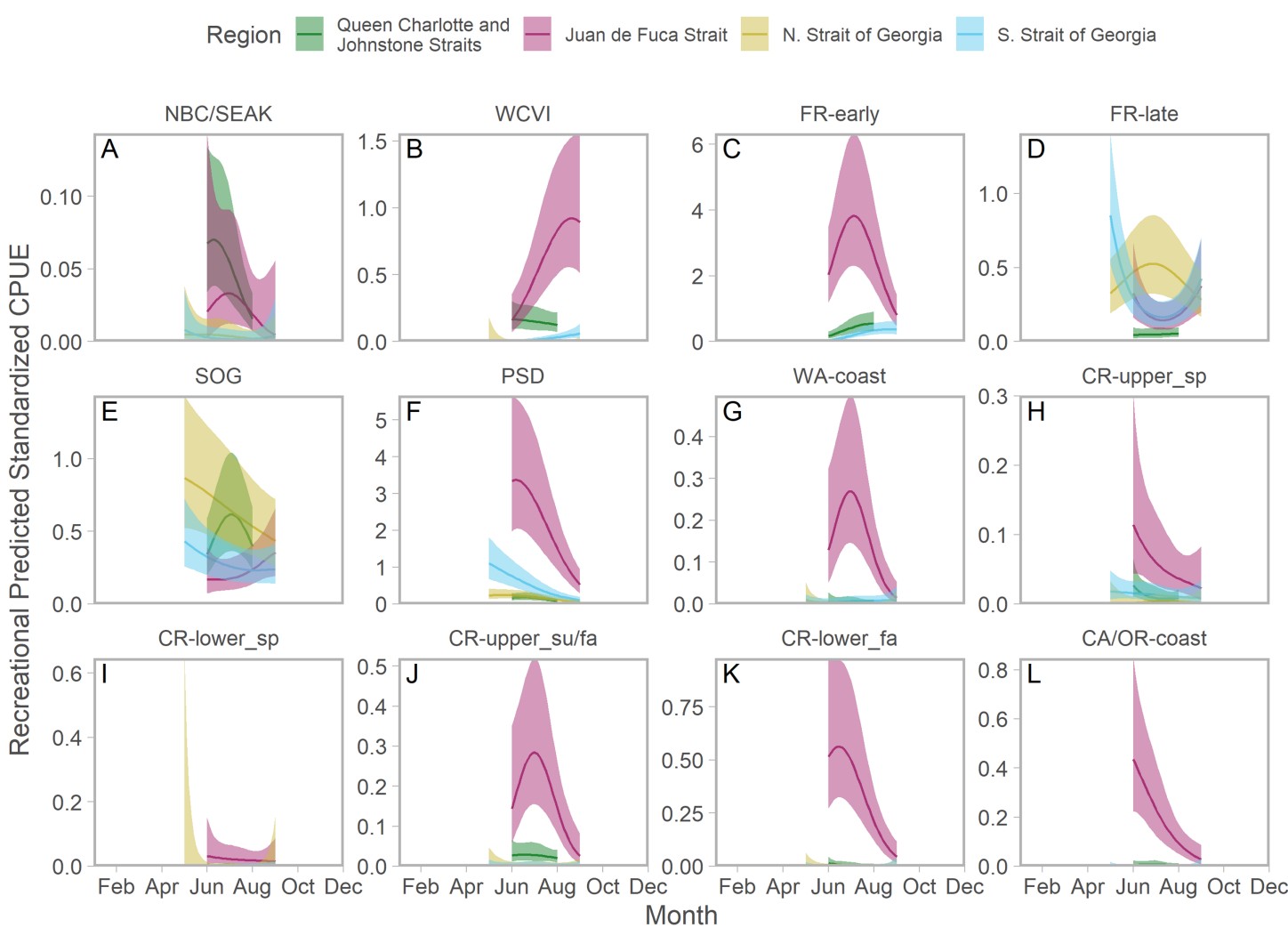

**Figure 5 Seasonal trends in model-predicted stock-specific standardized catch per unit effort (represents thousands of fish assuming fixed mean recreational effort) of regional stock aggregate in inside catch regions (colours) estimated using recreational fisheries data.** Ribbons represent 95% confidence intervals. Predictions are for an average year, integrating over annual random effects. Stock aggregates are arranged approximately latitudinally, based on freshwater entry point, from north to south (A–L). Note that y-axis scales differ among stock aggregates.

Subyearlings (i.e., fish that enter without overwintering as juveniles in freshwater) rear near their natal rivers throughout their first marine winter before migrating north and maturing along the continental shelf, while yearling Chinook salmon may migrate offshore immediately or after several months of shelf residence, depending on stock (*Healey, 1983*; *Trudel et al., 2009*; *Weitkamp, 2010*; *Tucker et al., 2012*; *Fisher et al., 2014*). While stock-specific distributions at fine spatial and temporal scales are often incorporated into local fisheries management decisions, these data may not be publicly available or widely disseminated (*DFO, 2019*). Here, we used extensive genetic sampling of southern BC fisheries to estimate seasonal trends in abundance and stock composition, and identified differences in marine migration behaviour among Chinook salmon populations. Similar to recent work on resident Chinook salmon within Puget Sound (e.g., *O'Neill & West,*

*2009*; *Chamberlin et al., 2011*), our results suggest a continuum of marine distributions or migration behaviours. Even among stocks spawning in close proximity and with similar freshwater life histories, we observed divergent patterns in the timing and extent to which stocks use nearshore habitats. Such differences in distribution may result in unique responses to basin-scale environmental drivers, predators, and fisheries.

## Regional variation

Although Chinook salmon standardized CPUE, a proxy for relative abundance, generally peaked in mid-summer, seasonal trends differed among regions. Broadly, this variation suggests Chinook salmon use habitats on the west coast of Vancouver Island differently to regions within the Salish Sea, though both fall within Healey's category of a continental shelf distribution (*Healey, 1983*). On the west coast of Vancouver Island, seasonal peaks in relative abundance were broad due to a mix of resident and migratory life-history types, as well as a particularly diverse stock composition. Conversely, the seasonal peak in relative abundance in Juan de Fuca Strait was compressed, consistent with the region being used predominantly as a migratory corridor for a smaller number of stocks returning to systems within the Salish Sea. The other inside regions (Queen Charlotte and Johnstone straits, as well as the northern and southern Strait of Georgia), exhibited much weaker seasonal cycles. The lack of an obvious seasonal peak in relative abundance in the southern Strait of Georgia, which had robust estimates of catch and effort throughout the year, suggests that substantial numbers of fish remain resident year-round or return to inside waters before recruiting to the fishery (45 or 62 cm fork length depending on PFMA). Chinook salmon residence has been relatively well documented in Puget Sound (*O'Neill & West, 2009*; *Chamberlin et al., 2011*; *Chamberlin & Quinn, 2014*; *Arostegui et al., 2017*); however, in the Strait of Georgia, this life-history strategy has not received extensive attention in the primary literature (but see *Healey & Groot, 1987*).

Coast-wide analyses indicate the stock composition of adult Chinook salmon varies spatially, with marine distributions correlated with freshwater life history (*Healey, 1983*; *Fisher et al., 2014*) and ocean entry location (*Weitkamp, 2010*). We found that each region's catch was dominated by three or four stock aggregates with resident populations eventually replaced by migratory stocks in summer and fall. Yet seasonal patterns in composition were highly variable among regions. Some of these patterns are intuitive (e.g., stocks that migrate directly from freshwater to the California Current are rarely observed within the Salish Sea). Others, however, are less obvious. For example, Puget Sound stocks were common in southern outside (SWVI) and inside (southern Strait of Georgia and Juan de Fuca Strait) regions during winter and spring, but in regions slightly further north they were replaced by similar stocks (i.e., predominantly fall run, subyearling) from the Columbia River basin or the Strait of Georgia.

## Stock-specific patterns

Patterns of stock-specific relative abundance (i.e., stock-specific standardized CPUE) within southern BC emphasize subtle differences in distribution and migratory behaviour among Chinook salmon populations that are lost when broadly categorizing stocks as

continental shelf residents or offshore migrants. For example, the standardized CPUE of many subyearling, fall run populations was greatest in NWVI (e.g., Strait of Georgia, upper Columbia River summer/fall, CA/OR coastal), but lower Columbia River and Fraser River late run populations were more abundant in SWVI. The relative difference in standardized CPUE between NWVI and SWVI also ranged from ambiguous (e.g., Puget Sound) to dramatic (e.g., Washington coastal). While ocean-type stocks that are observed in southeast Alaskan troll fisheries are often referred to as far-north migrating (*CTC, 2019*), our seasonal estimates of stock-specific abundance suggest differences in distribution may also occur further south. Such stock-specific patterns are likely a result of differences in marine maturation grounds, but may also be influenced by migratory behaviours, such as travel speed or distance from shore, that moderate a stock's exposure to fisheries.

Chinook salmon residence in southern continental shelf, and even nearshore, habitats during non-migratory periods is well known by fisheries managers and described qualitatively in Pacific salmon life history texts (e.g., *Riddell et al., 2018*). However, by quantifying seasonal trends in relative abundance, we are able to better resolve stock-specific distributions. First, Puget Sound fish were abundant throughout the year in both SWVI and NWVI, peaked in abundance in late winter or early spring in the southern Strait of Georgia, and migrated through Juan de Fuca Strait in late summer. This diversity may represent multiple allopatric components of the same cohort or perhaps movements between basins associated with ontogeny. Substantial numbers of Puget Sound Chinook salmon are resident within the sound during winter and early spring (*O'Neill & West, 2009*; *Chamberlin et al., 2011*; *Chamberlin & Quinn, 2014*), and individuals may exhibit restricted localized distributions (*Arostegui et al., 2017*).

Second, Columbia River (except lower spring run populations) and Strait of Georgia stocks were also present in NWVI during the winter and spring. Yet seasonal patterns in abundance were much more pronounced than in Puget Sound fish, suggesting the majority of each stock migrates north or offshore. Individuals caught in winter and spring may represent an early arrival by northern/offshore fish that will mature the following year or the southern portion of those populations' marine distribution.

Third, Fraser River late run (fall run, subyearling fish) and Strait of Georgia (predominantly fall run, subyearling east coast Vancouver Island populations) stocks were abundant early in the year in the Strait of Georgia. Yet there was also a late summer peak in the abundance of these stocks on the west coast of Vancouver Island, followed by an increase in Juan de Fuca Strait. These patterns suggest that Fraser River late run and Strait of Georgia stocks may exhibit multiple, distinct migratory behaviors. Some proportion appear to either return early to inside waters via Johnstone Strait (a migratory pulse that has not been well-resolved with available data) or else remain resident within the Salish Sea until they recruit into the fishery. Another component of these stocks appears to migrate from the continental shelf through Juan de Fuca Strait immediately prior to spawning migrations. If, as we suspect, some fish remain resident within the Salish Sea, there may be substantial implications for management. For example, previous estimates of poor

overwinter survival from acoustically tagged east coast Vancouver Island Chinook salmon may be biased low (*Neville, Beamish & Chittenden, 2014*).

## Conservation implications

Near year-round occupancy of the west coast of Vancouver Island and the Strait of Georgia by Chinook salmon emphasizes these locations should be considered foraging and maturation habitats for a relatively large number of stocks. Importantly, each region has oceanographic characteristics that differ from each other, as well as from offshore and northern areas (*Ware & McFarlane, 1989*; *Mackas & Coyle, 2005*). The west coast of Vancouver Island is the approximate location of the bifurcation of the North Pacific Current and straddles downwelling-dominated regions to the north and upwelling-dominated regions to the south (*Ware & McFarlane, 1989*). Interannual variation in the latitude of the bifurcation point and basin-scale climate forcing (e.g., the Pacific Decadal Oscillation) influences primary productivity and zooplankton community composition, with subsequent effects on higher trophic levels, including salmon (*Peterson, 2009*; *Sydeman et al., 2011*; *Malick et al., 2017*). Conversely, the Strait of Georgia is a protected coastal sea that is strongly influenced by estuarine circulation and freshwater inputs, rather than vertical transport (*LeBlond, 1983*). These traits have resulted in zooplankton communities (*Mackas et al., 2013*) and fish population dynamics (e.g., herring; *Cleary et al., 2020*) that are distinct from those of the continental shelf.

Distinct spatio-temporal distributions may contribute to variation among salmon stocks in growth and productivity. While declines in Chinook salmon body size and age-at-maturity are widespread, their extent varies among regions (*Ohlberger et al., 2018*; *Oke et al., 2020*). Given that Pacific salmon size- and age-at-maturity are influenced by growth late in marine residence (*Quinn, 2018*), stock-specific adult marine distributions may be essential to identifying mechanistic drivers of these declines. Already CWT recovery data suggest Chinook salmon may exhibit stock-specific shifts in marine distribution as climate change progresses, which could magnify divergent responses to what is commonly viewed as a shared environment (*Shelton et al., in press*). From a metapopulation perspective, variation in marine distributions may contribute to salmon portfolio effects and stabilize aggregate abundance (*Freshwater et al., 2018*). Indeed Chinook salmon populations entering the Salish Sea, a region that we have illustrated to contain greater variability in marine behaviour, exhibit weaker synchrony in productivity than stocks that enter the California Current or Gulf of Alaska (*Dorner, Catalano & Peterman, 2017*; *Ruff et al., 2017*).

Interspecific interactions, including predation, are likely impacted by differences in spatio-temporal distribution among Chinook salmon populations. Resident killer whales preferentially target Chinook salmon and early in the summer depend heavily on spring and summer run yearling Fraser River stocks (predominantly from the upper and middle portions of the watershed) (*Hanson et al., 2010*). Although the absolute abundance of yearling Fraser River Chinook salmon (all early run timing populations) is less than summer and fall run subyearling stocks (*CTC, 2019*), our estimates of relative abundance

suggest that yearling stocks may be particularly available to southern resident killer whales due to their compressed migration through Juan de Fuca Strait. Conversely, Fraser River fall run stocks have a more dispersed spatial and temporal distribution within southern BC, which may reduce their availability as prey.

## Data and model structure assumptions

Our predictions of stock-specific abundance depend on imperfect sampling of fisheries-dependent data, resulting in several assumptions. First, we accounted for annual variation via random effects. Such an approach was necessary due to imbalanced sampling across the annual cycle among years and relatively few samples for many strata; however, without year-specific fixed effects, the model will underestimate anomalous boom or bust years in stock-specific abundance. Second, estimates of catch and effort for the recreational fisheries data, derived from creel surveys and overflights, are particularly uncertain. Since this observation error was not incorporated into our predictions, the uncertainty associated with standardized CPUE estimates in inside regions is underestimated.

Third, it is unclear how precisely stock-specific harvest reflects stock-specific relative abundance because fisheries-independent estimates of mature Chinook salmon abundance in marine areas are not available. For example, catchability may vary seasonally due to changes in the depth distribution or behaviour of fish, as well as changes in fleet composition (e.g., highly motivated and skilled fishers may make up a greater proportion of the fleet during winter). Management actions, such as area closures and non-retention periods, will impact fisher behaviour and may also decouple catch from abundance. As a result, we can provide only minimum estimates of relative abundance of Fraser River early run stocks in Juan de Fuca Strait and the southern Strait of Georgia, as well as WCVI stocks in NWVI and SWVI, because spatio-temporal closures are used to minimize incidental harvest of those stocks (*DFO, 2012*; *Dobson, Holt & Davis, 2020*). Estimates of composition may be particularly biased in Juan de Fuca Strait and the southern Strait of Georgia, where recreational fisheries release relatively large numbers of individual that are difficult to genetically sample (*Dobson, Holt & Davis, 2020*; see Supporting Information for additional information). These and other difficulties associated with fisheries-dependent data emphasize the importance of developing robust catch sampling programs and of leveraging multiple data sources to inform fisheries management decisions.

Age structure is not incorporated into many salmon distribution models derived from GSI data (including the one presented here), but is a logical addition to increase ecological realism. CWT recoveries suggest older juvenile (*Trudel et al., 2009*) and adult (*Weitkamp, 2010*) Chinook salmon age classes, within a stock, are distributed further from their natal streams. GSI analyses incorporating age data from scale samples or associated CWT indicators could be used to resolve how robust this pattern is and increase the spatio-temporal resolution of movement models. Disentangling age- and stock-specific marine distributions may be necessary to identify the mechanisms driving declines in older age classes (*Ohlberger et al., 2018*; *Oke et al., 2020*).

Our model estimates stock-specific abundance as a function of total abundance and stock composition. Yet stock-specific abundance is commonly inferred directly from CWT recoveries, which are expanded for tagging and sampling effort, then used to calculate total abundance and stock composition as necessary. Such an approach approximates reality, where total abundance is the sum of stock-specific abundance, and is necessary for CWT recoveries because the entire catch is not tagged (thus composition estimates cannot be applied to total catch). We believe, however, that this method is less appropriate for GSI data because the expansion factors that are necessary to account for variable sampling effort when analyzing CWT recoveries are not readily available for GSI samples. We account for variable sampling effort by explicitly incorporating sample size into the stock composition component of the model, which has the additional benefit of accounting for uncertain stock assignments. Nevertheless, comparing inferences derived from CWT and GSI data using a common model, perhaps by estimating stock-specific abundance as latent variables constrained by total abundance, would be a valuable addition.

## CONCLUSIONS

We believe that seasonal patterns of composition and stock-specific abundance derived from GSI data can continue to improve fisheries management and our understanding of Chinook salmon marine ecology. Future work could integrate GSI information with data from other sources to clarify how ontogeny influences residence in different regions or compare the distributions of tagged and untagged stocks. Where composition and catch data can be matched at fine spatio-temporal scales, the model we present can be readily down-scaled and used to guide tactical fisheries management decisions. Similarly, by incorporating equivalent data on freshwater migration timing (*Parken et al., 2008*), this framework could be extended throughout the migratory corridor. Finally, predictions of seasonal abundance and composition could be used to parameterize simulation models to inform strategic decisions related to ecosystem-based fisheries management.

We used a novel statistical model integrating catch, effort, and genetic data to predict how stock-specific Chinook salmon abundance changes seasonally within nearshore regions of southern British Columbia. We found that stocks, even those with similar freshwater life histories and geographically proximate spawning locations, exhibited marked differences in whether and when they used specific nearshore habitats. These patterns represent distinct marine behaviours and may result in Chinook salmon exhibiting stock-specific responses to environmental drivers such as basin-scale oceanographic patterns or marine mammal abundance.

## ACKNOWLEDGEMENTS

We are grateful to DFO Southcoast Area staff and their partners who collected the catch, effort, and genetic stock composition data. Lee Kearey, Karin Mathias, and Bryan Rusch provided particularly helpful insight when interpreting data. We also thank participants of the Avid Anglers program who voluntarily submit biological data to improve salmon stock assessment. Will Duguid, James Losee, and Chuck Parken provided valuable information during discussions on Chinook salmon life history. Feedback from Kendra

Holt, Will Satterthwaite, and an anonymous reviewer greatly improved the quality of this manuscript.

### Funding
This work was supported by the Pacific Salmon Commission Southern Endowment Fund and Fisheries and Oceans Canada. The funders had no role in study design, data collection and analysis, decision to publish, or preparation of the manuscript.

### Grant Disclosures
The following grant information was disclosed by the authors:
Pacific Salmon Commission Southern Endowment Fund.
Fisheries and Oceans Canada.

### Competing Interests
The authors declare that they have no competing interests.

### Author Contributions
- Cameron Freshwater conceived and designed the experiments, performed the experiments, analyzed the data, prepared figures and/or tables, authored or reviewed drafts of the paper, and approved the final draft.
- Sean C. Anderson performed the experiments, analyzed the data, prepared figures and/or tables, authored or reviewed drafts of the paper, and approved the final draft.
- Terry D. Beacham performed the experiments, authored or reviewed drafts of the paper, and approved the final draft.
- Wilf Luedke performed the experiments, authored or reviewed drafts of the paper, and approved the final draft.
- Catarina Wor conceived and designed the experiments, authored or reviewed drafts of the paper, and approved the final draft.
- Jackie King conceived and designed the experiments, authored or reviewed drafts of the paper, and approved the final draft.

### Animal Ethics
The following information was supplied relating to ethical approvals (i.e., approving body and any reference numbers):

Fisheries and Oceans Canada provided approval for DFO staff to sample fish for scientific purposes under a blanket Section 52 license.

### Field Study Permissions
The following information was supplied relating to field study approvals (i.e., approving body and any reference numbers):

Fisheries and Oceans Canada provided approval for DFO staff to sample fish for scientific purposes under a blanket Section 52 license. (File Number 2018-502-0012).

## Data Availability

Data and code necessary to complete all analyses are available at
DOI 10.5281/zenodo.4672524.

## Supplemental Information

Supplemental information for this article can be found online at http://dx.doi.org/10.7717/
peerj.11163#supplemental-information.

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
