# Peer review of "An integrated model of seasonal changes in stock composition and abundance with an application to Chinook salmon"

_PeerJ, doi:10.7717/peerj.11163_

## Round 0.1 · original submission · Minor Revisions

Thank you very much for submitting this well written manuscript. As you will see below, both reviewers made some comments that they both think can be addressed with a minor revision. I agree and would ask you to please consider all reviewer comments and incorporate them into your manuscript where appropriate.

I just wanted to make a final point here but seconding one of reviewer 1's comments:
"All data are provided, as is the code, which is laudable."
Thank you very much for doing this. I'm a big supporter of this open and transparent way of doing science.

·

Basic reporting

Language is very clear and professional throughout. I have only very minor suggestions (though several), laid out below. The introduction and background are strong (again with some minor wording suggestions), the literature cited is appropriate, the structure is clear, and the figures are relevant and high quality. Raw data, along with code, are provided in a comprehensive GitHub repository, which is laudable.

Minor suggestions for text:

Line 37: May be appropriate to add a citation at end, or drop the language about "becom[ing] emblematic".

Line 49: As a practical matter, do ocean fisheries really operate "year-round" in these areas (winter fisheries are certainly rare coastwide, but perhaps less so in these areas)?

Line 50 and multiple times thereafter: I would suggest not using the term "mark" when referring to coded-wire tags (or to when later referring to fish being effectively marked by their genotypes), rather I would call both of these "tags". In the salmon management literature I am familiar with, "mark" seems generally reserved for externally visible, binary ways of distinguishing fish (e.g, adipose fin has either been clipped off or is present) and most commonly refers to the adipose fin clip, whereas a "tag" is often not externally visible and seemingly always encodes finer scale information (e.g. a CWT provides information on stock of origin, age, and often many other characteristics of the hatchery release group or wild fish tagging event it came from; even a FLOY tag conveys more than a simple binary 0/1 type of information) and most often refers to a CWT or to the genotype of a fish whose parents were also genotyped as part of a parentage based tagging program. In this context, the information provided by genotype in a GSI study seems much closer to a traditional "tag" than a traditional "mark". This would also be more consistent with how "mark" and "tag" are used on line 72, and when additional references are made later to mark-selective fisheries. However I realize things aren't always clear cut or terms used consistency, as highlighted by the fact pointed out on line 147 that the agency responsible for recovering coded-wire tags is called the Mark Recovery Program.

Line 62: replace "ress" with "in press" (or 2021 as appropriate)

Lines 72-73: While it is true that "mark-selective fisheries will result in differences in exploitation rates between marked indicator stocks and unmarked populations", the relevance here is not clear and I suggest dropping it (the part of this sentence about mass marking reducing the efficiency of CWT recovery is relevant and I suggest retaining that part). Although the authors do not directly suggest this, the complications posed by mark-selective fisheries are often a challenge to the CWT program that is brought up by advocates of replacing the CWT system with one based on genetics, implying that genetics can solve this particular problem. It is not obvious to me that genetics can do so -- the real problem is a lack of sampling of fish that are released rather than retained for potential sampling. In the absence of such sampling, a CWT-based approach may offer some advantages over a genetics-based approach when it comes to assessing mark-selective fisheries, since (despite problems in implementation) double-index tagging in combination with electronic tag detection may offer the best hopes for quantifying impacts of MSF, and it is not clear how electronic tag detection would work for genotypes. This is all quite aside from the main point of this paper, which is why I suggest simply dropping this clause.

Lines 75-76; Around here it is maybe worth more explicitly making the point (which is suggested earlier) that for "stocks" without tagging programs, GSI is the only way to get stock-specific information. On the other hand, is it worth discussing cases where genetic reporting groups may be more coarsely defined than some particular "stocks" of management interest (perhaps this is never an issue for the stocks of interest here, but is elsewhere in the range)?

Line 85: Though it's a bit hard to point to a direct line to a specific management action for either paper (or for Bellinger et al. 2015), Satterthwaite et al. 2014 (TAFS:143:117-133, http://dx.doi.org/10.1080/00028487.2013.837096) was probably more directly relevant to PFMC concerns and decision making than the cited 2015 paper.

Line 101: Are these tags "artificial" or just "physical" or "mechanical"?

Line 126: Presumably at least some (probably all to at least some extent) of these stocks are harvested in US fisheries as well as Canadian fisheries, even if you are not analyzing data from US fisheries?

Line 128: Can you provide a citation here that defines PFMAs?

Line 131-133: I'm not sure I understand what you are saying here. Area versus subarea has not been defined, I'm also not sure if PFMAs correspond to areas or subareas and whether catch regions are the same as areas, or an aggregation of areas. I'm assuming this is a case where managers place certain subareas in one larger area, when you think grouping them with a different area makes more geographic/ecological sense, but I'm not entirely clear on the wording and the hierarchy of scales.

Line 153: Was the contracted troller instructed to pursue their typical fishing strategy, and did they have incentives to maximize catch (e.g., pay per sample?). If fisheries in this area have minimum size limits or are mark-selective, was sampling done on all fish caught or only those which could have been retained in normal fishing?

Line 169: Are any guidelines provided on how to choose the one fish per day that is genetically sampled?

Line 175: Consider also denoting the "inside" regions on Figure 1.

Line 436: replace "ress" with "in press".

Figure S7: I would consider replacing "Composition" with "Proportion" on the y axis, since each panel only shows the proportions contributed by one stock rather than the total composition in an area (like Figure 3).

Figures in general: For figures with observational data plotted, consider whether approximate error bars could be added legibly (i.e. the error associated with individual observations, not just the model uncertainty ribbons).

Experimental design

This is original primary research with a well defined question, that addresses an identified knowledge gap (the distribution of many natural-origin Chinook stocks, and Chinook distributions at a finer patio-temporal scale than is usually studied. The work adheres to high technical and ethical standards.

The methods are generally described with sufficient detail and information to replicate, with the following exceptions:

Lines 203-207: Though the choices strike me as reasonable, can the authors say any more about how/why they chose these values of k?

In between lines 207 and 208 (not sure what happened to the line numbering here), more detail is needed on how the Dirichlet-multinomial allowed the model to "account for uncertainty in individual stock assignments" -- presumably it somehow made use of the posterior assignment probabilities coming out of cBAYES, but how? Especially since accounting for this uncertainty is highlighted in both the abstract (line 19) and introduction (line 116), it seems like more attention is warranted.

Somewhat similarly, the abstract (line 19) also highlights accounting for the uncertainty associated with annual variability in sampling effort, but this isn't really treated in much detail. Presumably, sampling effort relates to the total size of the n_j in equation 3a, and this leads to greater uncertainty when sampling is limited, but this isn't really made explicit. Also, it appears total catch C is assumed to be observed without error (equation 1a) but presumably there is some estimation involved in determining total catch for a sampling stratum, and the precision of this estimate probably also depends on sampling (though a different kind of sampling than the collection of tissues for genetic analysis). Finally, when calculating CPUE, uncertainty can be related to the total amount of effort, i.e. effort is itself a measure of how extensively the fishery has "sampled" the underlying abundance. A bit more clarity on which aspects of sampling effort were versus were not explicitly considered, and their consequences, might be informative.

Lines 211-212. I'm not entirely sure where the "very small values" were plugged in in place of zero "observations". It seems like the observations of zeros would be for certain n_j in equation 3a, in which case it seems like an integer value might be needed. Is it rather than some elements of theta (I wouldn't really call these observations) were fixed at 0.00001?

Validity of the findings

All data are provided, as is the code, which is laudable. The data are what they are, almost entirely collected from fishery-dependent surveys, thus there is no control (which isn't really meaningful for an observational study, but the reviewer guidelines ask about controls) and the statistical design makes good use of what is available. While the data are not the product of a designed experiment or even a sampling scheme optimized for the intended purpose, this is pretty much state of the art as far as salmon ocean distribution data go, and comparable too many published studies.

Conclusions are well stated, linked to original research questions, and the difference between direct results and speculation are clear.

I did wonder a bit about the approach of starting from modeling total (relative) abundance by month/area along with stock proportions by month/area rather than modeling stock-specific abundance by area and then the resultant summed abundance and stock proportions. Mechanistically, total abundance must be driven by the sum of each stock's abundance. Assuming that stocks are similarly distributed across years, as stocks vary in their relative abundance, it seems that the spatial pattern in total abundance across space must also vary, which conflicts somewhat with assuming a constant smooth pattern across years. Perhaps the random effects (in CPUE and/or in the proportions) serve to sufficiently soak this up, and this assumption is needed for model tractability, but I'd like to see just a bit more discussion/justification of this approach. This may just be a point to address in the Discussion -- what are the advantages of trying to directly model area-specific abundance of individual stocks, and from that derive stock proportions versus modeling a typical distribution of total abundance over space and typical stock proportions in each area? Maybe referencing Figures S7 and S9-S11 and what they imply about how constant or variable stock composition tended to be across years in a particular month/area would shine light on how important this choice of approach might be?

Lines 429-440: This would requite speculating outside the realm of model outputs, but would including age as a covariate in these models change any of the interpretations here, and/or would the potential that distributions are age-specific affect how we interpret the patterns presented in this paper (and in fairness, many related papers), where all ages are combined? (I see age effects are brought up later, around line 475)

Circling back to lines 207-208, it is stated that a "benefit" of the approach is "account[ing] for uncertainty in individual stock assignments", so perhaps somewhere there should be a mention of how much uncertainty there actually was in stock assignments, and whether this has any meaningful impact on the results or how they should be interpreted.

Lines 488-490 seem like important points. That said, if there are stock-specific responses to environmental drivers, and stocks vary across years in their relative abundance, what does this mean for the advisability of modeling total abundance across space separately from stock proportions versus modeling stock-specific abundance across space and deriving stock proportions from that? I am not suggesting that the authors should be required to develop an alternate model from this starting point, simply suggesting some further discussion of the issue and the choices made.

Additional comments

An impressive amount of work, clearly communicated. My suggestions are mostly minor attempts to improve the clarity of presentation and to give a bit more discussion to the motivation for certain modeling choices.

Reviewer 2 ·

Basic reporting

No comment

Experimental design

No comment

Validity of the findings

No comment

Additional comments

This is a review of Freshwater et al. I found the paper to be clear and well written, and appreciated the detail of the figures and supplemental material. Minor comments below -- my only hangup was not being able to install the software package.

- line 140: In the data collection section, I think it'd be helpful to add a little more detail about the exact effort you used from each set of samples (e.g. boat days or vessels for commercial & recreational), and what was used with the First Nations fishery [edit: this is boat days on ~ line 201, but would be helpful to discuss in this section]

- line 190: perhaps add a sentence similar to the hatchery / wild comparison, that the groupings in Table 1 also are lumping stocks with different marine distributions (e.g. spring & fall, Puget Sound & WA coast)

- line 194: how are the seasonal strata defined in the spatiotemporal strata? [edit: realizing you have this on 201]

- line 200: if these are overall pretty stationary, might be useful to include an AR coefficient 
- line 201: it might be interesting in future versions of this package to model the PFMA-specific smooths hierarchically (e.g. https://peerj.com/articles/6876/)

- line 208: I'm not used to seeing equation (3a) written this way and am maybe confusing myself. Might be easier to break it into 2 equations, with P ~ Multinomial(theta, N) and theta ~ Dirichlet (gamma)
- line 209: I think it'd be cool in future work to evaluate any interannual variation in the seasonal pattern (hierarchical smooths or year-specific smooths)
- line 263: I had trouble building the package across a couple computers -- there's some errors in the code that need fixing (https://api.travis-ci.org/v3/job/750093842/log.txt)

- line 268 / figure 2: One thing that's a little bit troubling is that the CPUE predictions / CIs are widest in summer months when you have the most data (though your Fig S1 shows some regions with little data in summer). I think this is an issue of the transformation from log to normal space, because Fig S3 shows that the predicted log-catch rates have symmetrical error bars 
- related to previous comment, I also wonder if it isn't possible to adjust the basis functions or GAM penalties (e.g. making the PFMA specific season smooths hierarchical)? Or using seasonal ARIMA models instead of basis functions? Though I guess with the latter the CIs will be symmetric in log space. 
- line 284: I wonder how much these trends for S. Strait of Georgia are affected by the PS blackmouth effect -- in winter/spring they may be one of the few stocks present, but are dominated by more migratory stocks in summer/fall [edit: I see this is in 1st discussion paragraph]. It'd be super interesting then to look at this composition over time + how it relates to Puget Sound hatchery practices

- Figs 3/4: nice work with these, I think they'll be seen as super useful

- line 478: or compare CWT distributions to GSI distributions for well-tagged stocks

---

## Round 0.2 · accepted · Accept

Both reviewers are happy with your revisions and so am I. Thank you very much!

Reviewer 1 had one comment for your consideration, but its entirely up to you if you want qualify or revise the statement they were referring to.

·

Basic reporting

Excellent.

Experimental design

Excellent.

Validity of the findings

Excellent.

Additional comments

The combination of revisions and responses to reviewer comments leave me completely satisfied that my comments and the comments of the other reviewer were appropriately addressed.

Just to prove I read the revised manuscript, one very minor quibble concerns line 114: Certainly Satterthwaite et al. 2015, confined to California, is in an area with lower stock diversity by most any reasonable standard. Satterthwaite et al. 2013 covers a broader geographic range (including most of Oregon) but only discusses 4 tagged stocks, so characterizing it as from a less diverse region or at least providing insight into a lower diversity of stocks is entirely appropriate. However, Bellinger et al. 2015, in their Figure 4, show results for 21 different reporting units. Some of the reporting units are quite coarse and include multiple "stocks" (e.g., "Alaska") but I would say 16 or so correspond to reasonably well defined "stocks" (and arguably, the very coarse nature of some reporting groups indicates even higher diversity). This contrasts with 11 "stocks" in Figure 3 of this manuscript. So it's not clear which study area is more "diverse", though diversity may not necessarily correspond to richness and richness/diversity per unit area or length of coastline covered might also factor into such a comparison. Overall I leave it to the authors to decide if that statement needs to be qualified or revised, and it is quite minor in the grand scheme of things.

Reviewer 2 ·

Basic reporting

Great - no issues

Experimental design

N/A

Validity of the findings

Great - no issues

Additional comments

I'm totally happy with all the revisions -- nice work overall with the paper!